# Microstructural Changes in the Spinothalamic Tract of CPSS Patients: Preliminary Results from a Single-Center Diffusion-Weighted Magnetic Resonance Imaging Study

**DOI:** 10.3390/brainsci13101370

**Published:** 2023-09-26

**Authors:** Richard L. Witkam, Lara S. Burmeister, Johan W. M. Van Goethem, Anja G. van der Kolk, Kris C. P. Vissers, Dylan J. H. A. Henssen

**Affiliations:** 1Department of Anaesthesiology, Pain and Palliative Medicine, Radboud University Medical Center, 6525 Nijmegen, The Netherlands; 2Department of Medical Imaging, Radboud University Medical Center, 6525 Nijmegen, The Netherlands; 3Department of Medical and Molecular Imaging, VITAZ, 9100 Sint-Niklaas, Belgium

**Keywords:** CPSS, DW-MRI, spinothalamic tract, microstructural change, biomarker

## Abstract

Introduction: Chronic pain after spinal surgery (CPSS), formerly known as failed back surgery syndrome, encompasses a variety of highly incapacitating chronic pain syndromes emerging after spinal surgery. The intractability of CPSS makes objective parameters that could aid classification and treatment essential. In this study, we investigated the use of cerebral diffusion-weighted magnetic resonance imaging. Methods: Cerebral 3T diffusion-weighted (DW-) MRI data from adult CPSS patients were assessed and compared with those of healthy controls matched by age and gender. Only imaging data without relevant artefacts or significant pathologies were included. Apparent diffusion coefficient (ADC) maps were calculated from the b0 and b1000 values using nonlinear regression. After skull stripping and affine registration of all imaging data, ADC values for fifteen anatomical regions were calculated and analyzed with independent samples T-tests. Results: A total of 32 subjects were included (sixteen CPSS patients and sixteen controls). The mean ADC value of the spinothalamic tract was found to be significantly higher in CPSS patients compared with in healthy controls (*p* = 0.013). The other anatomical regions did not show statistically different ADC values between the two groups. Conclusion: Our results suggest that patients suffering from CPSS are subject to microstructural changes, predominantly within the cerebral spinothalamic tract. Additional research could possibly lead to imaging biomarkers derived from ADC values in CPSS patients.

## 1. Introduction

Chronic pain after spinal surgery (CPSS) [1], previously known as ‘failed back surgery syndrome’ (FBSS) [2], or, more recently, by the term ‘persistent spinal pain syndrome’ [3], is an umbrella term for refractory chronic pain conditions after spinal surgery. Mostly, patients are diagnosed with CPSS if they suffer from recurrent or persistent back pain or radicular leg pain, but a combination of both could also be present. CPSS patients, especially those with neuropathic pain components, are highly disabled and disclose significantly worse levels of quality of life, pain and unemployment in comparison to those with other chronic pain syndromes [4,5].

Due to its multifactorial and still partially elusive etiology, CPSS is recognized as being challenging to treat [6,7] despite the availability of multiple treatment modalities [8]. Conventional therapies (e.g., pain medication) often provide insufficient pain relief [5]. When no treatable substrate behind the CPSS-related pain is identified either (e.g., imaging does not elucidate a recurring disc herniation), symptomatic interventional treatments such as spinal cord stimulation (SCS) can be considered. Although SCS has shown promising analgesic effects across multiple quality-of-life-related domains [7,9], it does not treat the underlying, unidentified cause of CPSS in that particular patient. Hence, the intractability of CPSS in such patients highlights the compelling need for more objective parameters. 

Imaging is one of the most robust methods that enables objective assessment of disease. Within neuropathic pain studies, perhaps the best-known companion imaging technique is magnetic resonance imaging (MRI). MR diffusion-weighted imaging (DW-MRI) has been used to assess microstructural integrity of cerebral regions [10]. This technique is based on the Brownian movement of water molecules and measures the extent of diffusion in a particular anatomical region or volume, where it uses additional gradient pulses with certain magnitudes in orthogonal directions to measure the net displacement of these water molecules. The strength and timing of such gradient pulses, and, with these, the degree of diffusion weighting, are determined by the so-called b-values [10]. In general, the higher the b-value, the more profound the diffusion weighting effect will be. After image acquisition, the apparent diffusion coefficient (ADC) for each voxel can be calculated, ultimately resulting in an ADC map which reflects the extent of the water diffusivity of different regions of interest [11].

In general, lower ADC values indicate more restricted diffusion, which is associated with higher tissue density and a more organized microstructure. Conversely, higher ADC values indicate less restricted diffusion, which is associated with lower tissue density and is an indication of a loss of microstructural organization [12]. Similarly, intact axonal membranes are considered to be the key determinants of anisotropic water diffusion in white matter [10], while gray matter predominantly shows isotropic diffusion. DW-MRI is used daily in clinical radiological practice, and the derived ADC values have been shown to strongly correlate with tissue integrity, especially in patients suffering from cerebral ischemia [13]. For CPSS patients in particular, measuring ADC values might point towards certain areas in which the neuronal structures might have undergone alterations, possibly by means of processes related to neuroplasticity. To name a few hypothetically generated examples, ADC values could elucidate loss of neuronal tissues due to degradation or an edematous state due to inflammation. Subsequently, the corresponding affected areas or neuronal circuitries can be used to sharpen our current knowledge on the etiopathogenesis of CPSS, especially when imaging is performed during different stages of CPSS. In addition, any objective imaging substrates derived from DW-MRI would also be valuable in patients suffering from CPSS in whom the underlying cause could not be elucidated. Such substrates might serve as possible objective biomarkers in pain patients. Pain is known as a subjective sensory experience which can be influenced by various physiological and psychological factors. The search for more objective biomarkers could provide an improved understanding of the pathophysiological mechanisms involved in the development of chronic pain. This, in turn, could lead to a paradigm shift in the way by which different chronic pain syndromes are defined and diagnosed and may lead to scrutiny with regard to the current definitions and diagnostic approaches to chronic pain syndromes [14,15]. In addition to providing more insights into pathophysiological mechanisms and helping to re-define diagnostic criteria, the role of objective biomarkers in pain patients as predictive features will become increasingly potent as well. When more targeted therapeutic agents or targets are discovered, the use of objective biomarkers will become paramount as reproducible patient selection based on phenotypical characteristics is key [15,16]. Thereby, objective biomarkers of pain can then be used as companion diagnostic tools to ensure that patients receive the most optimal treatment for their respective chronic pain syndromes [17]. Subsequently, the same biomarkers may then also be used as an objective follow-up tool to determine the response to a specific treatment. In general, the field studying objective pain biomarkers can be divided into two distinct groups. First, the research groups investigating physiological pain biomarkers have focused on genetics, vesicular micro-RNA and metabolic and molecular pathways, as well as stress-related biomarkers, in patients suffering from chronic pain. Second, the search for neuroimaging biomarkers in chronic pain patients initially investigated the functional activity of a wide variety of brain regions [18,19], although molecular imaging methods will allow for in vivo evaluation of neurochemical networks and specific receptors [20]. 

In this study, we contribute to our understanding of neuroimaging biomarkers in patients suffering from CPSS. Since the integrity of neuronal tissues might be altered or become compromised after prolonged exposure to stimuli, as is the case within chronic pain patients, we explored the use of DW-MRI to investigate any possibly present microstructural cerebral changes in patients suffering from CPSS. 

## 2. Methods

### 2.1. Ethical Approval

Ethical approval was waived by our institutional review board due to the retrospective nature of this study. Patients who chose—before their clinical MRI examinations—to opt out of sharing their anonymized clinically acquired data for scientific purposes were excluded. 

### 2.2. Study Population 

By means of a cross section based on diagnosis and disease codes, all medical records of adult CPSS patients (≥18 years) treated at the Radboud University Medical Center were identified and screened. Patients were eligible for inclusion when they reported only sensory complaints and there was availability of an MRI examination of the brain containing a DW-MRI sequence for that patient. In addition, the MR imaging of the patient’s brain had to be free from relevant imaging artefacts (e.g., susceptibility artefacts from dental implants) or significant brain pathology (e.g., brain infarction, diffuse leukoaraiosis or neuro-oncological disease). 

As controls, patients without low back or leg pain were included; this control cohort consisted of patients who underwent an MRI examination for a suspected unruptured cerebral aneurysm observed through CT imaging. When MR imaging showed no cerebral aneurysm or other brain pathology or relevant imaging artefacts, the patient was included as a control subject. These subjects were matched to the included CPSS patients based on age and gender.

### 2.3. MRI Protocol

All MRI examinations had been performed on a 3T Siemens MRI platform (i.e., Prisma-Fit and Skyra) using protocols developed for clinical purposes; for this study, only the T_1_-weighted and diffusion-weighted MRI data from these protocols were used. The T_1_-weighted images were acquired using a 3D T_1_-weighted MPRAGE sequence (repetition time (TR), 2100 ms; echo time (TE), 2.42 ms; spatial resolution, 1 mm isotropic). The DW-MRI consisted of a diffusion-weighted sequence with two b-values (b0 and b1000) with a TR of 4210 ms, TE of 75 ms and a spatial resolution of 5 × 5 × 5.5 mm^3^. The ADC maps were automatically calculated from the b0 and b1000 values using nonlinear regression. 

### 2.4. Data Processing

The age, gender and pain distribution patterns of the included participants were extracted from the medical records. MRI data were anonymized and extracted from the picture archive and communication system. The T_1_-weighted MRI was used for registration purposes. Prior to registration, skull stripping was carried out using the Brain Extraction Tool from the software library of the Oxford Centre for Functional MRI of the Brain (FMRIB) [21]. The MNI152 linear dataset of the Montreal Neurological Institute (MNI) was used as a registration database. This dataset comprised 152 normal T_1_-weighted images with a 1 mm isotropic voxel size which were linearly co-registered using a 9-parameter algorithm to the Talairach coordinate system, available from the Human Connectome Project library [22,23]. The MNI152 linear dataset was skull stripped, and segmentation of the remaining structures was carried out using the methodology mentioned before. The data of each subject were linearly registered using the FMRIB Linear Image Registration Tool (FLIRT) algorithm using the FSL toolbox. This resulted in a registration matrix of each individual subject for the MNI152 dataset. This registration matrix was subsequently used to robustly transform the ADC map into the MNI152 dataset. 

Segmentations of anatomical atlases registered in the MNI152 space using the Talairach coordinate system were extracted. Segmentations of cortical and subcortical structures were extracted from the Human Brain Project atlas, while those of white matter tracts were extracted from the HCP1065 dataset. Segmentations of projection white matter tracts (e.g., corticospinal tract and spinothalamic tract) included both the infratentorial and supratentorial portions. The segmentations of various structures and white matter tracts were overlaid on the ADC map of each subject. The quality of the segmentations was visually assessed by one of the researchers (D.H., >8 years of experience with experimental neuroimaging). Poorly fitting segmentations were manually corrected (e.g., subtle cortical atrophy, ex vacuo enlargement of the ventricular system). The data handling processes are summarized in Figure 1.

Mean ADC values of each segmented brain region were calculated for each individual subject. The included regions of interest comprised: (1) white matter (total volume of subcortical white matter); (2) cortical gray matter; (3) cerebellum (both gray and white matter, including the deep cerebellar nuclei); (4) thalamus; (5) putamen; (6) globus pallidus externa; (7) globus pallidus interna; (8) caudate nucleus; (9) amygdala; (10) hippocampus; (11) precentral gyrus; (12) postcentral gyrus; (13) insula; (14) corticothalamic tract; (15) spinothalamic tract. 

### 2.5. Statistical Analysis

All calculated ADC values were exported to SPSS (IBM Corp (2017). IBM SPSS Statistics for Windows, Version 27.0. Armonk, NY, USA: IBM Corp.) for further statistical analysis. As most of the data were normally distributed according to the Shapiro–Wilk test, independent samples T-tests were performed to test for statistical significance between the ADC values of the CPSS patients and those of the healthy controls for all segmented anatomical regions. Statistical significance was assumed when *p* < 0.05. 

## 3. Results 

In this study, sixteen CPSS patients were included. As each patient was age and gender matched with a healthy control, a total of 32 subjects were included. Both groups comprised eight males, and the average age was 58.1 (±9.1) and 60.2 (±13.3) years in the CPSS and control groups, respectively. Due to the presence of both right-sided and left-sided pain in the included CPSS patients, the splitting of results into distinct left- and right-sided components was considered unviable. 

The average ADC values (mean ± SD) of the evaluated anatomical structures of both groups, together with the calculated *p*-values, are depicted in Table 1. Voxel-wise comparison showed that the mean ADC value of the spinothalamic tract was significantly higher in CPSS patients (*p* = 0.013). The ADC values of the other white matter tracts were not significantly different between CPSS patients and healthy controls. Additionally, voxel-wise comparison of cortical and subcortical gray matter structures revealed no significant differences between both groups. 

## 4. Discussion

This study shows that patients suffering from CPSS had higher mean ADC values within the cerebral spinothalamic tract as compared to controls. No other brain structures were found to have a significantly different mean ADC. These results suggest that microstructural changes may have occurred within the cerebral spinothalamic tract of CPSS patients. The exact biochemical or cellular processes leading to these microstructural changes, however, remain elusive. An increased ADC might result from loss of membrane integrity, leading to a larger extracellular compartment within a voxel. Within white matter, an increased diffusivity can be observed in the case of axonal degeneration, neuronal loss or demyelination [24]. It has also been described that microstructural damage to the spinothalamic tract results in decreased diffusivity in a plethora of pain syndromes [25]. However, whether these microstructural changes underpin the observed changes in diffusivity is still unknown. Regarding the involvement of the spinothalamic tracts within the pain circuitry, it is generally accepted that these bundles play a central role in conveying noxious stimuli, which are processed by the cerebral cortex as painful. That these white matter tracts also play a role in the development of chronic neuropathic pain disorders is highlighted by various diffusion tensor tractography studies [26,27,28]. In the study of Yoon et al., however, the spinothalamic tract showed no significant differences in patients suffering from neuropathic pain after spinal cord injury as compared to in healthy controls [29]. They reported decreased diffusivity in parts of the corticospinal and thalamocortical tract, as well as in the splenium, the body of the corpus callosum and the right superior longitudinal fasciculus. The authors hypothesized that a neuroinflammatory state might explain the decreased diffusivity. In a state of neuroinflammation, microglia within the sensory neuraxis might undergo morphological changes. The activated microglia possess larger cell bodies with thicker and shorter processes as compared to normal microglia. Additionally, microglia are usually found in increased numbers in neuroinflammation [30]. These changes result in a decreased volume of the extracellular compartment, leading to decreased diffusivity. However, as we found an increased diffusivity within the spinothalamic tract, the role of neuroinflammation in CPSS patients must be further investigated. A limitation of the study of Yoon et al., however, is formed by the mixed motor and sensory complaints which occur in patients suffering from spinal cord injury. Thereby, it is not clear which neurological deficit drives microstructural changes in different white matter bundles [29]. Another paper investigating neuropathic pain after spinal cord injury also disclosed mixed changes in diffusivity across gray matter structures (i.e., parts of the medial and lateral pain matrix and of the reward circuitry of the brain). In the paper of Gustin et al., the white matter tracts between the gray matter structures were not found to undergo microstructural changes [31]. Tractography results did, on the other hand, reveal that the posterior parietal cortex projected to most gray matter structures, which revealed anatomical changes. Although the role of the posterior parietal cortex in the processing of chronic neuropathic pain remains poorly understood, it has been suggested that it modulates the thalamic output and influences the overall activity of regions of the pain matrix [31]. The results from the present study, however, do not corroborate the microstructural changes of cortical and subcortical gray matter described by Gustin et al. 

The novelty of the current study lies within its assessment of the microstructural integrity of the brains of CPSS patients by use of DW-MRI imaging. The use of robust and well-defined anatomical atlases as source material for segmentations is considered a strength. However, the retrospective study design and the limited study population are relevant impediments which prevent the drawing of definite conclusions. Another limitation concerns the relatively low spatial resolution of the DW-MRI sequence in relation to the finer anatomical structures assessed in this study by means of the higher-resolution T_1_-MPRAGE. Therefore, this study can be perceived as hypothesis generating. 

The clinical implementation of DR-MWI for CPSS patients across multiple centers might face some difficulties. Although the radiological equipment is probably available in most hospitals or pain clinics, standardized imaging protocols are lacking, which can be considered a key factor for obtaining future robust imaging datasets. A consensus on both data acquisition and analysis methods should be established, as the ADC values derived from DW-MRI could differ across various scanners [32]. Other issues that might occur, when not appropriately optimized, were summarized by deSouza et al. (e.g., a low signal-to-noise-ratio of data, imaging distortion, motion artefacts and varying data processing techniques) [33]. Furthermore, the presence of multiple partially understood influencing factors such as patient age, duration of pain, pain distribution pattern and the stage or moment in time regarding CPSS might alter imaging outcomes and, with that, hamper the development of reliable imaging protocols. Overall, addressing these factors holds multiple future challenges, both for clinicians and industries. Expert meetings or consortia may become crucial to play a mediating role between these parties [33].

Future studies should hold a greater number of subjects and should use more sophisticated imaging techniques to investigate microstructural changes more effectively. In addition to yielding high-resolution images at stronger magnetic field strengths, we suggest the use of diffusion kurtosis imaging. Additionally, as neuroinflammatory processes could contribute to the described microstructural changes, quantitative sensory mapping, magnetic resonance spectroscopy and ultrasmall-paramagnetic-iron-oxide-nanoparticles-enhanced neuroimaging could be considered. 

## 5. Conclusions

Our study suggests that microstructural changes occur within the cerebral spinothalamic tract of patients suffering from CPSS. However, the underlying mechanisms remain unknown. 

## Figures and Tables

**Figure 1 brainsci-13-01370-f001:**
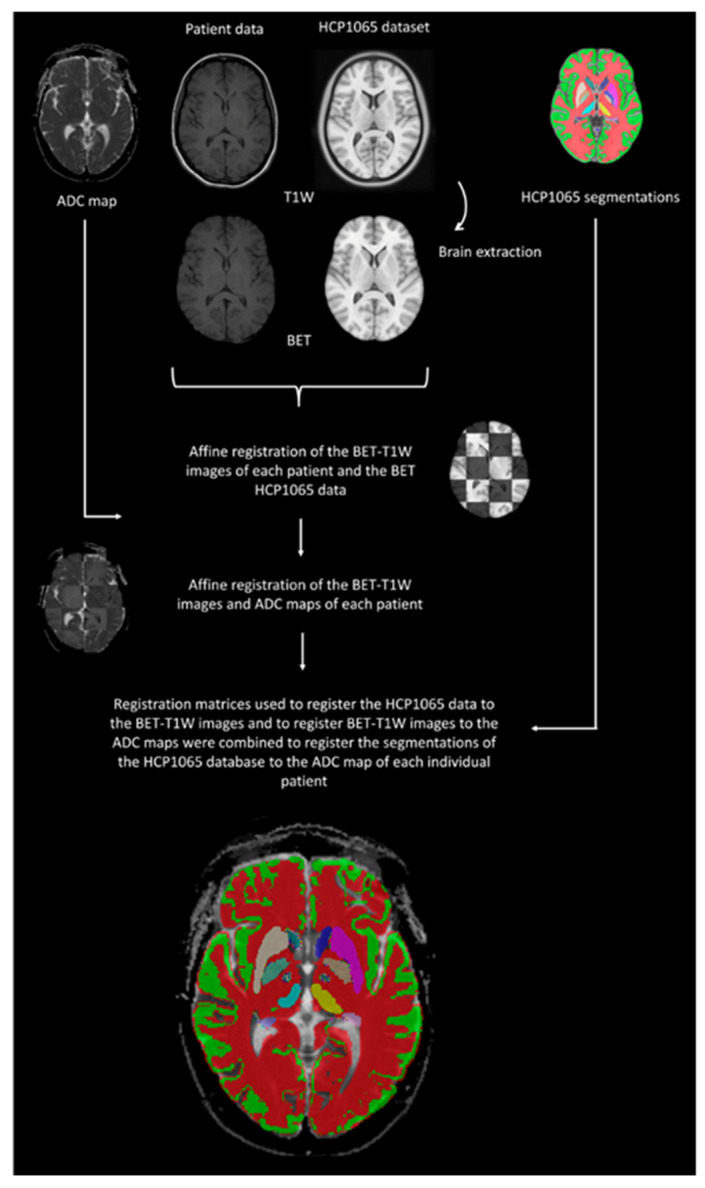
Overview of the data handling process. Legend: BET, Brain Extraction Tool; HCP, Human Connectome Project library. First, all imaging data were stripped from non-brain tissue through the BET. Subsequently, the data were affinely registered using a linear image registration tool. For each patient and control, the ADC maps were aligned with the segmentations of the studied anatomical regions, from which the corresponding ADC values were calculated.

**Table 1 brainsci-13-01370-t001:** Outline of the average ADC values.

	CPSS Group (Mean ± SD)	Healthy Controls (Mean ± SD)	*p*-Value
White matter	966.2 ± 71.7	1005.6 ± 105.8	0.227
Gray matter (cortical)	1090 ± 87.4	1163.4 ± 123.3	0.061
Cerebellum	874.6 ± 84	862.4 ± 76.4	0.671
Thalamus	983 ± 218.3	896.4 ± 139.1	0.191
Putamen	815.1 ± 45.3	835.3 ± 101.5	0.474
Globus pallidus externa	949.9 ± 94.3	997.2 ± 163.9	0.325
Globus pallidus interna	1071.9 ± 183.1	1138.2 ± 242.7	0.390
Caudate nucleus	1262.6 ± 422.4	1320.5 ± 500.8	0.726
Amygdala	950.4 ± 94.9	952.4 ± 169.4	0.969
Hippocampus	969.9 ± 88.4	952.7 ± 135.3	0.674
Precentral gyrus	1109.8 ± 62.7	1145.5 ± 119.2	0.301
Postcentral gyrus	1127 ± 71.3	1155.2 ± 128.4	0.449
Insula	1035.1 ± 110.6	1040 ± 139.8	0.912
Corticothalamic tract	901.7 ± 95.6	893.3 ± 110.9	0.820
Spinothalamic tract	922.8 ± 130.9	826.6 ± 52.2	**0.013**

Legend: CPSS, chronic pain after spinal surgery. Both groups consisted of 16 patients. The white matter contained all cerebral white matter regions or structures (i.e., including the other white matter structures which were also segmented separately).

## Data Availability

The data presented in this study are available on request from the corresponding author.

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
