# Peer review of "Microstructural Changes in the Spinothalamic Tract of CPSS Patients: Preliminary Results from a Single-Center Diffusion-Weighted Magnetic Resonance Imaging Study"

_brainsci, 2023, doi:10.3390/brainsci13101370_

Round 1

Reviewer 1 Report

Very well presented and scientifically supported manuscript,  based on the preliminary results that were extracted from the imaging findings, regarding the microstructural changes in the spinothalamic tract of CPSS patients. The scientific soundness and the originality of the study are important and the overall merit for the readers is high. The scientific documentation and the statistical analysis that accompanied the results of the study are adequate in order to support the conclusions of the study. However, as this study was based on a relatively restricted population of patients, I would recommend that the title of the manuscript should be modified in order to include the term 'preliminary results from a single-center study'.  

Author Response

Please find our answers within the attached file

Reviewer 2 Report

The Authors present the results of patients with microstructural changes, predominantly within the cerebral spinothalamic resultant from the chronic pain after spinal surgery. This research could lead to imaging biomarkers derived from apparent diffusion coefficient map values in

patients’ chronic pain after spinal surgery. The topic is interesting and seems to be important bearing in mind the need to properly diagnose and treat many patients that undergo spinal surgery complications. The experiments were designed and conducted via interesting ways and performed in accordance with a dedicated method.

Question 1: I suggest the authors to briefly describe about the importance of ADC values in the introduction section of the manuscript.

Question 2:  Are there any other reported methods to evaluate microstructural changes, If yes please explain how the method adopted by the authors is more beneficial over the previously reported methods in the discussion section.

Question 3: Please briefly describe the limitations with the proposed method for implementation in clinic and what further research direction of research is necessary to implement the proposed method for patients.

Author Response

Please find our answers within the attached file.
